

# Next-generation radiance unfiltering process for the Clouds and Earth's Radiant Energy System instrument

Lusheng Liang[1], Wenying Su[2], Sergio Sejas[1], Zachary Eitzen[1] and Norman G. Loeb[2]

[1]Science Systems and Applications Inc, Hampton, Virginia, VA

[2]MS420, NASA Langley Research Center, Hampton, Virginia, USA

*Correspondence to*: Lusheng Liang (lusheng.liang@nasa.gov)

**Abstract.** The filtered radiances measured by the Clouds and the Earth's Radiant Energy System (CERES) instruments are converted to shortwave (SW), longwave (LW), and window unfiltered radiances based on regressions developed from theoretical radiative transfer simulations to relate filtered and unfiltered radiances. This paper describes an update to the existing Edition 4
CERES unfiltering algorithm (Loeb et al., 2001), incorporating the most recent developments in radiative transfer modeling, ancillary input datasets, and increased computational and storage capabilities during the past 20 years. Simulations are performed with MODTRAN 5.4. Over land and snow, the surface Bidirectional Reflectance Distribution Function (BRDF) is characterized by a kernel-based representation in the simulations, instead of the Lambertian surface used in the Edition 4 unfiltering process. Radiance unfiltering is explicitly separated into 4 seasonally dependent land surface groups based on the spectral radiation
similarities of different surface types (defined by International Geosphere-Biosphere Programme); over snow, it is separated into fresh snow, permanent snow, and sea ice. It contrasts to the Edition 4 unfiltering process that one set of regressions for land and snow, respectively.

The instantaneous unfiltering errors are estimated with independent test cases generated from radiative transfer simulations in
which the 'true' unfiltered radiances from radiative transfer simulations are compared with the unfiltered radiances calculated from the regressions. Overall, the relative errors are mostly within ±0.5% for SW, within ±0.2% for daytime LW, and within ±0.1% for nighttime LW for both CERES Terra Flight Model 1 (FM1) and Aqua FM3 instruments. The unfiltered radiances are converted to fluxes and compared to CERES Edition 4 fluxes. The global mean instantaneous fluxes for Aqua FM3 are reduced by less than 0.42 Wm$^{-2}$ for SW and increased by less than 0.47 Wm$^{-2}$ for daytime LW; for Terra FM1, they are reduced by less than 0.31 Wm$^{-2}$ for SW and increased by less than 0.29 Wm$^{-2}$ for daytime LW, though regional differences can be as large as 2.0 Wm$^{-2}$. Nighttime
LW flux differences are nearly negligible for both instruments.

## 1 Introduction

The Clouds and the Earth's Radiant Energy System (CERES) instruments have been continuously monitoring the earth's radiation budget at the surface, within the atmosphere, and at the top-of-atmosphere since 2000 (Wielicki et al. 1996). Currently, there are
six CERES instruments onboard four satellites observing the Earth: Flight Model 1 (FM1) and FM2 on the EOS Terra satellite since 1999, FM3 and FM4 on the Aqua satellite since 2002, FM5 on the Suomi National Polar-orbiting Partnership (NPP) satellite since 2011, and most recently, FM6 on the NOAA-20 satellite since 2017. CERES instruments measure radiances in shortwave (SW, 0.3 – 5 μm), window (WN, 8 – 12μm) and total (0.3 – 200 μm) channels for FM1-5 and SW, longwave (LW, 5 – 40 μm) and total channels for FM6. The reflected and emitted radiances from earth scenes enter the instrument aperture, pass through the
optical systems and are recorded by the instrument detectors and electronics [Loeb et al. 2001]. However, the measured filtered radiances must be converted to unfiltered radiances which are equivalent to the radiances arriving at the instrument prior to entering its optical system. The unfiltered radiances can be further converted to fluxes with angular distribution models (ADMs, Su et al. 2015) for scientific research [e.g. Sherwood et al. 2018, Loeb et al. 2021].

The radiance unfiltering process in the CERES Edition 4 product uses theoretical radiative transfer simulations to construct
regression relationships between filtered and unfiltered radiances [Loeb et al. 2001]. If the CERES spectral response functions were spectrally invariant or earth scenes are spectrally invariant from each other, one set of regression coefficients would be sufficient to convert filtered radiances to unfiltered radiances. However, the spectral response functions are not spectrally flat (Fig. 1) and the reflected and emitted radiant energy is spectrally different among different Earth targets (Fig. 2). The radiance unfiltering process must therefore be scene-type dependent. In the CERES Edition 4 radiance unfiltering process, regression coefficients were
constructed for land, ocean, and snow/sea ice, and further separated into clear and cloudy cases.

A reliable relationship between filtered and unfiltered radiances preferably requires accurate spectral simulations cover a wide range of earth-atmosphere conditions . With the advances in radiative transfer models, increased computational power and storage,



and advances in new observations of earth-atmosphere during the past 20 years, we are able to update the CERES radiance unfiltering process. For radiative transfer modeling, MODRTRAN 3.7 is replaced by MODTRAN 5.4 with HITRAN database updates for atmospheric gas absorptions and incorporating of discrete-ordinate-method radiative transfer (DISORT, Stamnes et al., 1997) to calculate multiple scattering faster and with higher fidelity. Over land, the simulations in the Edition 4 radiance unfiltering process used Lambertian surfaces for a limited number of surface types, which might not represent various land surface types sufficiently; Furthermore, one set of regression coefficients were developed and applied regardless of the spectral differences among land surfaces. With the advancement of land surface albedo/BRDF observations from the Moderate Resolution Imaging Spectroradiometer (MODIS), we are able to characterize surface the surface Bidirectional Reflectance Distribution Functions (BRDFs) much better than the Lambertian representation in the model simulations. It allows us to identify spectral radiation characteristics among various land surface types, both globally and temporally, upon which the unfiltering coefficients can be developed and applied to more specific land surface types.

To further reduce unfiltered radiance uncertainties, we also use simulations that better match observations to develop the regression coefficients. Over ocean, a better implementation of the Cox-Munk BRDF model [Cox and Munk 1954] is used. Over land, we modify the BRDF kernel parameters from MODIS BRDF/albedo products to better capture the hot spot feature for vegetation [Maignan et al., 2004]. Over snow/seaice, simulations that best match to observations are used to develop the regression coefficients for Greenland and Antarctica for permanent snow, seaice, and fresh snow, respectively. For overcast simulations, we replace the built-in cloud properties in MODTRAN 5.4 with realistic ones.

In addition to the above improvements, we increase the number of solar zenith angle (SZA) and viewing zenith angle (VZA) bins, at which to calculate the regression coefficients, to reduce the unfiltered radiance uncertainties. The number of SZA bins is increased from 5 to 13, VZA bins is increased from 5 to 6; the number of relative azimuth angle (RAZ) bins remains unchanged at 5 bins.

This paper is organized as follows. Section 2.1 describes the unfiltering algorithm, which is the same as that in Loeb et al. (2001). Detailed radiative transfer simulations are described in section 2.2. Section 3 presents error analysis of the unfiltering process. Section 4 presents applications of the updated radiance unfiltering process to CERES observed filtered radiances to obtain unfiltered radiances, with which the fluxes are converted for the four seasonal months in 2010.

## 2 Methodology

### 2.1 Unfiltering algorithm

The reflected solar and emitted thermal radiances from Earth's surface and atmosphere pass through the optics of CERES instruments. The filtered measurements must be converted to reflect the true reflected and emitted radiances prior to entering CERES instruments. The algorithms were described in detail in Loeb et al. (2001); a brief description is given below.

The unfiltered reflected shortwave, emitted longwave and window radiances are defined as follows:

$$m_u^{SW} = \int_0^\infty I_\lambda^r \, d\lambda \qquad (1a)$$
$$m_u^{LW} = \int_0^\infty I_\lambda^e \, d\lambda \qquad (1b)$$
$$m_u^{WN} = \int_{\lambda_1}^{\lambda_2} I_\lambda^e \, d\lambda \qquad (1c)$$

where $\lambda$ is the wavelength in um, $I_\lambda^r$ and $I_\lambda^e$ (Wm$^{-2}$sr$^{-1}$μm$^{-1}$) are the reflected solar and emitted thermal radiances, $\lambda_1$=8.1 μm and $\lambda_2 = 11.8$ μm define a wavelength interval within the thermal window range in CERES FM1-FM5. Given instrument spectral response functions, CERES measured filtered radiances can be modeled as

$$m_f^j = \int_0^\infty S_\lambda^j I_\lambda \, d\lambda \qquad (2)$$

where $S_\lambda^j$ is the spectral response function, $I_\lambda$ is the radiance incident on the instrument, j denotes the SW, TOT, or WN channel.

For CERES FM1-FM5, the relationship between filtered radiance and unfiltered radiance are constructed through regression as follows:

$$m_u^{SW} = a_0 + a_1 m_f^{SW_r} + a_2 \left(m_f^{SW_r}\right)^2 \qquad (3a)$$
$$m_u^{WN} = b_0 + b_1 m_f^{WN} + b_2 \left(m_f^{WN}\right)^2 \qquad (3b)$$
$$m_u^{LW}(DAY) = c_0 + c_1 m_f^{SW_r} + c_2 m_f^{TOT} + c_3 m_f^{WN} \qquad (3c)$$
$$m_u^{LW}(NIGHT) = d_0 + d_1 m_f^{TOT} + d_2 m_f^{WN} \qquad (3d)$$



For FM6, the window channel is replaced by a LW channel, and the unfiltered LW radiances can be determined by $m_f^{SWr}$ and $m_f^{TOT}$ during daytime, and $m_f^{TOT}$ at night:

$$m_u^{LW_{SW\_TOT}}(DAY) = e_0 + e_1 m_f^{SWr} + e_2 m_f^{TOT} \qquad (3e)$$

$$m_u^{LW_{SW\_TOT}}(NIGHT) = f_0 + f_1 m_f^{TOT} \qquad (3f)$$

The unfiltered LW radiances can also be determined directly from $m_f^{LW}$:

$$m_u^{LW_{LW}} = g_0 + g_1 m_f^{LW} + g_2 \left(m_f^{LW}\right)^2 \qquad (3g)$$

In Equations (3a)-(3g), $a_0$, $a_1$, $a_2$, $b_0$, $b_1$, $b_2$, $c_0$, $c_1$, $c_2$, $c_3$, $d_o$, $d_1$, $d_2$, $e_o$, $e_1$, $e_2$, $e_3$, $f_o$, $f_1$, $f_2$, $g_o$, $g_1$, and $g_2$ are theoretically derived regression coefficients. $m_f^{SWr}$ is the reflected portion of the filtered SW radiance and it is determined by removing the emitted thermal portion $m_f^{SW_e}$ from $m_f^{SW}$:

$$m_f^{SWr} = m_f^{SW} - m_f^{SW_e} \qquad (4)$$

For FM1-FM5, $m_f^{SW_e}$ is determined by a relationship between measured $m_f^{SW}$ and $m_f^{WN}$ at night:

$$m_f^{SW_e} = h_0 + h_1 m_f^{WN} + h_2 \left(m_f^{WN}\right)^2 \qquad (5a)$$

For FM6, $m_f^{SW_e}$ is determined by a relationship between measured $m_f^{SW}$ and $m_f^{LW}$ at night:

$$m_f^{SW_e} = k_0 + k_1 m_f^{LW} + k_2 \left(m_f^{LW}\right)^2 \qquad (5b)$$

### 2.2 Spectral radiance simulations

#### 2.2.1 Radiative transfer model

The regression coefficients used to convert CERES observed filtered radiances to unfiltered radiances are developed from radiative transfer simulations over typical earth scenes. We use MODTRAN 5.4 [Berk et al., 2016] for the radiative transfer simulations, replacing MODTRAN 3.7 in the Edition 4 unfiltering process. A few major improvements in MODTRAN 5.4 as compared to MODTRAN 3.7 include, but are not limited to [Berk 2004, Berk 2016]:

(1) MODTRAN 5.4 is based on HITRAN 2012 [Rothman, et al., 2013], while MODTRAN 3.7 is based on HITRAN 1992 [Rothman, et al., 1992],
(2) MODTRAN 5.4 uses correlated-k algorithm, which significantly improves the accuracy of multiple scattering calculations [Berk et al., 1998],
(3) MODTRAN 5.4 allows finer spectral resolution simulations,
(4) MODTRAN 5.4 fully treats the auxiliary molecular species, and
(5) MODTRAN 5.4 calculates multiple scattering faster and with higher fidelity with an improved incorporation of discrete-ordinate-method radiative transfer (DISORT, Stamnes et al., 1997) into MODTRAN 5.4.

The simulations are performed for clear and overcast conditions, and radiances for broken clouds are linearly weighted with clear-sky and overcast radiances for cloud fractions of 0.25, 0.50 and 0.75. Simulations are performed from 0.25 mm to 1000 mm with a spectral resolution of 2 wavenumbers per cm.

#### 2.2.2 Resolution in the number of angular bins

In the Edition 4 unfiltering process, regression coefficients are evaluated at 5 SZA bins, 5 VZA bins and 5 RAZ bins. The number of SZA bins is increased from 5 to 13, the number of VZA bins is increased from 5 to 6 to minimize the radiance unfiltering uncertainties, and the number of RAZ bins remains the same at 5 bins as in the Edition 4 unfiltering process. The detailed analysis will be shown in section 3. Table 1 shows the SZAs, VZAs, and RAZs at which the radiance unfiltering coefficients are determined for SW, daytime LW and WN radiances. For nighttime LW and WN radiances, the unfiltering regression coefficients are evaluated at VZA bins only.

#### 2.2.3 Clear-sky simulations over ocean

Over ocean, the surface is characterized by the Cox-Munk model [Cox and Munk, 1954] in the radiative transfer model simulations. The implementation of Takashima (1985) in MODTRAN 3.7 simulations for the Edition 4 CERES radiance unfiltering is replaced



by the Second Simulations of the Satellite Signal in the Solar Spectrum (6S) radiative transfer code [Vermote et al., 1997], which includes the radiance contributions from ocean whitecaps and underwater in addition to the specular reflections from the ocean surface. Figure 3 shows that the simulation by the 6S radiative transfer code characterizes the ocean surface radiances better than that of Takashima, particularly for VZAs greater than 50º.

The radiative transfer simulations over clear-sky ocean use the same atmospheric and surface temperature conditions as that in the Edition 4 (Table 2). For each solar-viewing angular bin, 7 simulations are used to calculate coefficients over clear-sky ocean.

**2.2.4 Clear-sky simulations over land**

The radiance unfiltering process is highly scene-dependent, and therefore, it is critical to identify and classify the scene types. In the CERES Edition 4 radiance unfiltering process, regression coefficients were constructed for land, ocean, and snow/sea ice, and further separated into clear and cloudy cases. Particularly, over land, Edition 4 used simulations for 6 land surface types, namely desert, dry sand, vegetation, coniferous forest, forest conifer species, and dry meadows grass (Table 4 in Loeb et al, 2001), from 155 which one set of coefficients was developed for each sun-viewing angular bin regardless of land surface types.

During the last 20 years, we have gained better observations of land surfaces. One of the observations is MODIS derived land surface albedo/BRDF products. It is based on the semiempirical reciprocal RossThick-LiSparse model [Li and Strahler, 1992; Lucht et al., 2000]. The BRF at the surface can be modeled as a linear combination of three terms:

$$\rho(\theta_0, \theta, \phi) = f_{iso} + f_{vol}K_{vol}(\theta_0, \theta, \phi) + f_{geo}K_{geo}(\theta_0, \theta, \phi) \quad (6)$$

where the first term on the right-hand side of the equation is the isotropic scattering contribution, $K_{vol}$ in the second term is the RossThick kernel to characterize volumetric scattering from horizontally homogeneous leaf canopies, and $K_{geo}$ in the third term is the LiSparse kernel to characterize geometrical-optical surface scattering from three-dimensional objects. The kernel fitting parameters $f_{iso}$, $f_{vol}$, and $f_{geo}$ were derived from atmospherically corrected, multi-angular land surface BRFs. The derived kernel 165 parameters are available at the 7 MODIS spectral bands (0.47 mm, 0.55 mm, 0.65 mm, 0.86 mm, 1.2 mm, 1.6 mm, and 2.1mm) over a 16-day cycle over land. Validation efforts have showed that the MODIS albedo/BRDF retrievals are in good agreement with field measurements, typically within 10% [Liang et al., 2002, Jin et al., 2003a, 2003b; Wang et al., 2004].

With the MODIS derived kernel fitting parameters $f_{iso}$, $f_{vol}$, and $f_{geo}$ from albedo/BRDF products for the 7 MODIS bands, we 170 are able to estimate the spectral fitting parameters, which are used to calculate the spectral radiations. The determination of the fitting parameters at other wavelengths across the SW and LW range is described as follows:
(1) at a wavelength between 0.47 µm and 2.1 µm, calculate the fitting parameters with all 7 band parameters by the spline interpolation.

(2) at a wavelength below 0.47 µm, the fitting parameter $f_\lambda$ is calculated based on the fitting parameters at 0.47 µm ($f_{0.47}$,)
and 0.55 µm ($f_{0.55}$), respectively, along with the spectral reflectances in JPL surface spectral reflectances. Equation 7 shows that one fitting parameter at $\lambda$ is estimated from $f_{0.47}$ by scalling it with $R_\lambda$ and $R_{0.47}$, which are the JPL surface spectral reflectances at $\lambda$ and 0.47 µm, respectively. Equation 8 shows that another one is estimated from $f_{0.55}$ scaled with $R_\lambda$ and $R_{0.55}$:

$$f_{0.47}^{est} = f_{0.47}\frac{R_\lambda}{R_{0.47}} \quad \text{and} \quad (7)$$
$$f_{0.55}^{est} = f_{0.55}\frac{R_\lambda}{R_{0.55}} \; . \quad (8)$$

The fitting parameter at $\lambda$ is then calculated as:
$$f_\lambda = f_{0.47}^{est}\frac{(\lambda-0.55um)^2}{(\lambda-0.47um)^2+(\lambda-0.55um)^2} + f_{0.55}^{est}\frac{(\lambda-0.47um)^2}{(\lambda-0.47um)^2+(\lambda-0.55um)^2}, \quad (9)$$
that is, we put more weights on $f_{0.47}^{est}$ than $f_{0.55}^{est}$.
(3) the same approach is used to calculate the fitting parameters at wavelengths above 2.11 µm based on the fitting parameters 185 at the 1.6 µm and 2.1 µm channels.

Figure 4 shows an example comparing SW reflected radiance simulations to CERES observed SW radiances for evergreen needleleaf forest. It clearly shows that the simulations with MODIS derived surface BRDF match observations far more better than the simulations with a Lambertian surface. However, the simulations based on MODIS retrieved kernel fitting parameters still 190 underestimate BRFs around the hot spot angles for vegetated surfaces, where the VZA is equal to the SZA in the backward direction.



Based on the RossThick-LiSparse model, Maignan et al. (2004) modified $K_{geo}$, the geometrical scattering kernel, to highlight the hot spot feature for vegetations. In this paper, we replaced $K_{geo}$ for RossThick-LiSparse model with that defined in Maignan et al. (2004). The calculation of new fitting parameters $f_{iso}$, $f_{vol}$, and $f_{geo}$ is described as follows:

195         (1) calculate BRFs at various sun-viewing angles with the MODIS retrieved fitting parameters for the 7 MODIS spectral bands. The calculations are performed with a bin width of 10° for SZA, VZA, and RAZ.

        (2) with the calculated BRFs in (1), calculate the fitting parameters based on the model described in Maignan et al. (2004) to highlight the hot spot feature of vegetations. The same model is also implemented in MODTRAN 5.4 to simulate the land surface reflectance.

Figure 4 shows that the simulations with the modified $K_{geo}$ capture the sharp increases of BRFs around the hot spot angles better than the simulations based on the original RossThick-LiSparse model.

More importantly, based on the spectral simulations with the MODIS derived kernel fitting parameters, we can identify the radiation spectral characteristics across various land surface types. Figure 5 shows the simulations for the 16 land surface types

defined by International Geosphere-Biosphere Programme (IGBP, Table 3) in January. It suggests that the spectral shapes differences among different surface types should be considered in the radiance unfiltering process. With the K-means clustering approach, the 16 simulations are classified into 4 groups in January based on their spectral similarities. Furthermore, although Fig. 6a indicates that the spectral shapes of evergreen broadleaf forest (IGBP 02) are similar in the four seasons, the spectral shapes of deciduous needleleaf forest (IGBP 03, Fig. 6b) are not, suggesting that we also need to consider the seasonal variations. Therefore,

for other seasons (April, July, and October), the grouping is different as compared to January (Table 4).

In a summary, the radiative transfer simulations over land are performed for each IGBP land surface type based on the 10-year averaged RosThick-LiSpase model fitting parameters from collection 6 MODIS derived albedo/BRDF products MCD43C1 [Strahler, 1999; Gao et. al., 2005]. The unfiltering regression coefficients for land are constructed for 4 surface groups in each of the 4 seasons (winter, spring, summer and fall, respectively). The surface temperatures are prescribed using the median values of

a 5-year surface temperature climatology for each IGBP type as calculated from the Goddard Earth Observing System reanalysis (Rienecker et al., 2008), version 5.4.1, included in the Edition 4 CERES Single Satellite Footprint TOA/Surface Fluxes and Clouds (SSF) product data. Depending on the location of a surface type, either a standard, midlatitude summer/winter, or subarctic summer/winter atmospheric profile is used. Dust aerosol is used over the bare soil and rocks (IGBP type 16) and open shrublands (IGBP type 7). Rural aerosol is used over other IGBP types. Depending on the surface types, the aerosol optical depths (AODs)

use 5, 25, 50, 75 and 95 percentiles of a 5-year AOD climatology from AOD retrieved by MODIS (collection 5.1), also included in the CERES SSF product. Simulations are also separated for daytime and nighttime with different surface temperature to account for diurnal temperature variations. For each land surface type, there are 8 to 10 clear-sky cases for each sun-viewing geometry.

### 2.2.5 Simulations over snow

It is still a challenge to simulate radiation from snow/ice surfaces. In the Edition 4 CERES radiance unfiltering process, the

simulations over snow surfaces were characterized by the Warren-Wiscombe model [Wiscombe and Warren, 1980]. Compared to CERES observations (Fig. 7), the simulations with the Warren-Wiscombe snow model overestimate BRF for smaller VZAs and underestimate for larger VZAs, whereas the simulations with the RossThick-LiSparse model are better although they are still unable to match observations at larger VZAs. The unfiltering regression coefficients is developed separately for permanent snow, fresh snow, and sea ice, which contrasts to that in the Edition 4 where one set of regression coefficients is used for snow and sea

ice . We select the best simulations to match the observations to develop regression coefficients to reduce the unfiltered radiance uncertainties as much as possible. The CERES clear-sky observations are compared to simulations based on 10-year of MODIS retrieved BRDF fitting parameters for Greenland and Antarctica using averages in April, July, October, and December, and all months. From these 10 simulation candidates, the SW regression coefficients are constructed from two simulations that best match and envelop observations for each snow/sea ice surface. For example, the regression coefficients over Greenland are calculated by

using simulations based on MODIS BRDF fitting parameters averaged over all months for Greenland and Antarctica; the regression coefficients over fresh snow is calculated by using simulations based on MODIS BRDF fitting parameter averages over Greenland in October and over Antarctica in April. Median values of surface temperature from a 5-year climatology for each snow/sea ice surface are used in the simulations for SW radiance unfiltering. Also from these 10 simulation candidates, the LW and WN regression coefficients are constructed from a simulation that best matches observations along with 3 surface temperatures which

are the 25th, 50th, and 75th percentile values of a 5-year climatology for each snow/sea ice surface. Tropospheric aerosols are used with a visibility of 300 KM and the subarctic winter atmospheric profile is used.

### 2.2.6 Simulations for overcast conditions

In the Edition 4 radiance unfiltering process, simulations for overcast conditions were performed with built-in cloud optical single scattering properties in MODTRAN 5.4, such as asymmetry factors, scattering coefficients and single scattering albedos. We





update them with more realistic cloud optical single scattering properties to better match the observed radiances. For water clouds, the single scattering properties, including phase functions, are based on the Mie scattering calculations; for ice clouds, a two-habit ice cloud model is used [Liu et al., 2016, Loeb et al., 2018]. As expected, the new simulations are able to capture the cloud anisotropic characteristics better. The overcast properties used in radiative transfer simulations over ocean, land and snow are shown in Table 5.

The simulations of deep convective clouds are also included to construct the LW regression coefficients over ocean and land. The conditions used in the deep convective cloud simulation are shown in Table 6.

### 2.2.7 Summary of constructed coefficients

To briefly summarize, the regression coefficients are calculated for 13 SZAs, 6 VZAs, and 5 RAZs for SW and daytime LW, and 6 VZAs for nighttime LW for each scene type. Scene type is determined by ocean, land (separated into 4 groups in each of the seasons: spring, summer, fall, and winter) and snow/sea ice (separated into permanent snow over Greenland and Antarctica, fresh snow, and sea ice). Over ocean and land, the coefficients for SW radiance unfiltering are derived separately for clear and cloudy conditions; over snow and sea ice, clear and cloudy scenes use the same coefficients. For LW and WN radiance unfiltering, one set of coefficients is used regardless of cloud coverage conditions. The coefficients derived from both clear and cloudy conditions are used if a scene lacks information of cloud coverage during the application.

## 3 Error analysis


The instantaneous errors in the unfiltered radiance are estimated by the same approach used in Loeb et al. (2001). A set of simulations (described in the following sub-sections) that differ from those used to construct the regression coefficients is generated, and it is assumed that they represent the true unfiltered radiances. The simulated radiances are convolved with CERES spectral response functions to obtain filtered radiances. The unfiltering regression coefficients are then applied to the filtered 265 radiances to get unfiltered radiances and they are compared to the "true" simulated radiances. In the following discussion, without explicitly stating, the errors are evaluated at 9 SZAs, 6 VZAs and 5 RAZs (Table 7) in daytime and 6 VZAs and 5 RAZs at nighttime. The following shows the error analysis based on CERES Terra FM1. The error analysis for Aqua FM3 can be found in the supplemental figures (Fig. S1-S10).

### 3.1 Resolution in the number of angular bins

In the Edition 4 radiance unfiltering process, the regression coefficients were evaluated at 5 SZAs, 5 VZAs, and 5 RAZs. For SZAs, they are 0º, 41.4º, 60.0º, 75.5º, and 85.0º. To evaluate if the 5 SZAs is sufficient, we used the simulations for clear-sky with the same conditions to generate the regression coefficients but at different SZAs: 29º, 51.3º, 68º, and 80.3º. Figure 8 checks if the unfiltered radiance errors at the testing SZAs is comparable to the regression errors for SZAs at 0º, 41.4º, 60.0º, 75.5º, and 85.0º. The larger errors in the test cases suggest that the number of SZAs used in Edition 4 is insufficient. We increase the number of 275 SZAs from 5 to 13 (Table 1) and the same approach is used to evaluate if it is enough. Figure 9 confirms that further increasing the number of SZAs is unnecessary. For the number of VZA and RAZ, we verify that 6 VZAs (increase one VZA at 70º) and 5 RAZs are sufficient. For the regression coefficients of overcast conditions, the evaluation results are similar to that for the clear-sky conditions.

### 3.2 Errors due to wind speed over ocean

As mentioned in Section 2, the regression coefficients for ocean are generated using radiative transfer simulations with a wind speed of 5 m/s, and they will be applied to oceanic scenes regardless of wind speed. Figure 10 shows that the SW unfiltered radiance errors for clear-sky scenes with wind speeds of 2 m/s and 12 m/s are comparable to the errors of the regression coefficients built with the wind speed of 5 m/s. It suggests that constructing unfiltering coefficients based on simulations with only a wind speed of 5 m/s is sufficient.

### 3.3 Errors due to aerosols for clear-sky scenes


Over ocean, maritime aerosols are used in radiative transfer simulations to generate regression coefficients (Table 2). We applied these regression coefficients to simulations with different aerosols for clear sky over ocean. Figure 11 shows the SW unfiltered radiance errors for dust and urban aerosols, and urban aerosol with larger AOD. Compared to the PDF of SW unfiltered radiance errors for scenes with maritime aerosols, the PDFs of errors for other aerosols become broader, and the PDF modes for dust and 290 urban aerosols are shifted to opposite directions from one another. The mean biases are within ±0.35% and RMS error are below 0.47%. Over land, depending on the surface type, dust or rural aerosols are used in simulations to generate regression coefficients.



Figure 12 shows the SW unfiltered radiance errors for scenes with larger AOD aerosols (varying from 0.5 to 2.0 depending on surface type, representing the 99th percentile in AOD climatology for each type) are larger than the scenes used to construct regression coefficients for land. As expected, the magnitude of errors become larger. The mean biases are within ±0.14% and RMS error are below 0.20%. Overall, errors due to aerosols for most scenes are within ±0.5%.

### 3.4 Errors due to scene identification

The CERES radiance unfiltering process is scene-type dependent. Due to cloud mask uncertainties, a clear-sky scene may be mistakenly identified as a cloudy scene, and vice versa. Therefore, an actual clear-sky scene may use regression coefficients for cloudy scenes, and a cloudy scene may use regression coefficients for clear-sky scenes. Taking the simulations to derive the unfiltering regression coefficients of clear-sky scenes over ocean and land in July, Fig. 13 shows that the PDF of SW unfiltered radiance errors for clear-sky scenes by using the regression coefficients derived from cloudy scenes is wider than that based upon regression coefficients for clear-sky scenes. Overall, the errors are within ±0.5%. The larger PDF difference over ocean than land is due to a larger contrast in reflectivity between clouds and the underlying surface over ocean than land. Alternatively, an overcast scene with thin clouds or a broken cloudy scene might be identified as a clear-sky scene. Taking the simulations to derive regression coefficients for cloudy scenes over ocean and land in July, Fig. 14 compares PDFs of SW unfiltered radiance errors for overcast scenes of cirrus with a cloud optical depth (COD) of 2 by using the regression coefficients derived from clear-sky scenes and cloudy sky scenes. As expected, the errors for scenes become larger if its corresponding regression coefficients are not used. Most scenes are still within ±0.5%, although the absolute errors can be as large as 1.0%. Figure 15 also compares PDFs of radiance errors for broken cloudy scenes with a cloud fraction of 10% by using the regression coefficients derived from clear-sky scenes. It shows that the PDFs of radiance errors are comparable with the mean errors near zero and RMS errors are less than 0.18%.

As discussed in Section 2, the regression coefficients over land are developed for 4 land surface groups (defined by IGBP types with similar SW spectral shapes) for each of the 4 seasons (Fig. 5 and Table 4). Here we further evaluate the SW unfiltered radiance errors for each land surface group by using the regression coefficients derived from a different land surface group. For example, Fig. 16a shows that the SW unfiltered radiance errors in January for Group 1, which contains land IGBP surface types 01, 02, 04, 05, 08, 12, and 14, caused by using the regression coefficients derived from all 4 groups. As expected, the errors would be smaller if the regression coefficients derived from its group are used, but larger otherwise. Particularly, the errors for Group 4 (Fig. 6d), which contains IGBP surface type 16, can be easily greater than 1.0% when the regression coefficients derived from other groups are used. Therefore, it is important to differentiate land surface types when developing the regression coefficients.

### 3.5 Errors for cloudy scenes

Four cloud properties over land and ocean are used in radiative transfer simulations of overcast scenes to derive regression coefficients for cloudy scenes (Table 5). One of the simulations represents stratus overcast scenes with a COD of 5.6. Simulations with the same condition but using a different COD of 38 are used to evaluate the SW unfiltered radiance errors. Figure 17 shows that the errors are similar to the regression errors for stratus overcast scenes with a COD of 5.6. The mean biases are within ±0.04% and RMS errors are below 0.10% over both ocean and land.

### 3.6 Errors for LW

As mentioned in Section 2, to derive regression coefficients for LW, the surface temperatures varying from 280 to 320 K over ocean are used in simulations; over land, the median values of surface temperature in a 5-year climatologies are used; and over snow/sea ice, the 25th, 50th, and 75th percentile values are used. For clear-sky conditions, we use simulations with the minimum and maximum surface temperatures from climatologies, keeping all other conditions the same to evaluate the unfiltered LW radiance error. Figure 18 shows that the errors are nearly the same as the regression errors. For overcast conditions, the unfiltered radiance errors are evaluated from simulations with clouds placed at different altitudes (Table 8) as compared to that used to generate the regression coefficients (Table 5). Figure 19 shows that the errors for overcast sky are not sensitive to the altitudes where clouds are placed with nearly negligible biases and RMS errors.

## 4 Impact of unfiltering algorithm on instantaneous fluxes

The newly updated regression coefficients are applied to the filtered radiances to obtain the unfiltered SW, LW and WN radiances of CERES Terra FM1 and Aqua FM3 instruments. With CERES Edition 4 ADMs (Su et al. 2015), the unfiltered radiances are converted to corresponding instantaneous fluxes. Figures 20 to 22 show the SW, daytime and nighttime LW flux differences between the newly calculated fluxes and the CERES Edition 4 fluxes for Aqua FM3 in January, April, July, and October in 2010 (the corresponding analyses for Terra FM1 are shown in Fig. S11 to S13).



For SW fluxes (Fig. 20), the regional differences are mostly negative over ocean and positive over snow. For deserts, the differences are positive except for Sahara and Middle East in January. For other snow-free land surfaces, the differences are mostly negative. The magnitude of regional differences can be larger than 2.0 Wm$^{-2}$. The global mean instantaneous fluxes are reduced by less than 0.42 Wm$^{-2}$. For Terra FM1 (Fig. S11), the differences show similar regional patterns as to Aqua FM3 but with smaller magnitudes with the global mean instantaneous flux reduced by less than 0.47 Wm$^{-2}$.

For daytime LW fluxes (Fig. 21), the differences are positive over ocean and negative over snow; over snow-free land surfaces, the differences are mostly positive except over Australia in January. The global mean fluxes are increased by less than 0.31 Wm$^{-2}$. For Terra FM1 (Fig. S12), the regional differences are mostly positive and smaller than that of Aqua FM3, with an overall global mean flux are increased by less than 0.29 Wm$^{-2}$.

For nighttime LW fluxes (Fig. 22), the differences are nearly negligible, and the global mean differences are less than 0.01 Wm$^{-2}$ for both instruments.

## 5 Summary

CERES instruments measure filtered reflected solar and emitted thermal infrared radiances from the earth-atmosphere. For use in science applications, the filtered radiances must be converted to unfiltered radiances, which are equivalent to the radiances arriving at the instrument prior to entering its optical system. The unfiltered radiances are then converted to radiative fluxes for scientific research. This paper describes an update to the existing Edition 4 CERES unfiltering algorithm (Loeb et al., 2001) by incorporating the most recent developments in radiative transfer modeling, ancillary input datasets, and increased computational and storage capabilities during the past 20 years. A few of the improvements in the new version are:

(1) Simulations are performed with MODTRAN 5.4 with many updates as compared to MODTRAN 3.7.
(2) Over ocean, the implementation of the Cox-Munk BRDF model in the 6S radiative transfer code replaces the implementation in Takashima (1985) used in simulations for the Edition 4 radiance unfiltering process. The newer version matches CERES observed SW radiances better.
(3) For simulations over land and snow, surface BRDFs are characterized by MODIS retrieved RossThick-LiSparse kernel-based BRDF model fitting parameters for each IGBP surface type, instead of the Lambertian surface used in simulations for the Edition 4 CERES radiance process. The hot spot features for vegetation are further modeled by using the approach described in Maignan et al. [2004].
(4) Over land, unfiltering regression coefficients are derived separately into 4 surface groups to characterize spectral differences among different surface types. The regression coefficients are also separated into 4 seasons to characterize the seasonal variation of the surfaces.
(5) The regression coefficients are calculated at more SZA bins, increased to 13 from 5 in SZA as used in the Edition 4, to reduce the unfiltered radiance errors.
(6) Climatological surface temperatures from Goddard Earth Observing System reanalysis were used in simulations over land, snow, and sea ice. Climatological AODs derived from MODIS over land are also used in the simulations.

Instantaneous unfiltered radiance errors were estimated using radiative transfer simulations. The simulated filtered radiances are converted to unfiltered radiances and compared to the simulated unfiltered radiances. Overall, the instantaneous relative errors are mostly within ±0.5% for SW radiances, within ±0.2% for daytime LW radiances, and negligible for nighttime LW and WN radiances.

The unfiltered radiances with the newly updated unfiltering regression coefficients are converted to fluxes and compared to fluxes in Edition 4. The global mean instantaneous fluxes for Aqua FM3 are reduced by less than 0.42 Wm$^{-2}$ for SW and are increased by less than 0.47 Wm$^{-2}$ for daytime LW; while for Terra FM1, the global mean instantaneous fluxes are reduced by less than 0.31 Wm$^{-2}$ for SW and increased by less than 0.29 Wm$^{-2}$ for daytime LW, though the regional differences can be as large as 2.0 Wm$^{-2}$. For nighttime LW fluxes, the differences are negligible for both instruments.

## Author contribution:

NL initialized and implemented the previous edition of unfiltering algorithm. LL and WS designed the algorithm improvements with contributions from SS and ZE. LL carried model simulations and implemented algorithm. LL, WS, SS, and ZE conducted analysis and prepared the manuscript. All authors contributed to the reviewing of the manuscript.



**Acknowledgments:** This research has been supported by the NASA CERES project. The MODIS MCD43C1 product was obtained from NASA Earthdata searching and ordering web tool (https://earthdata.nasa.gov) provided by NASA's Distributed Active Archive Center.

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




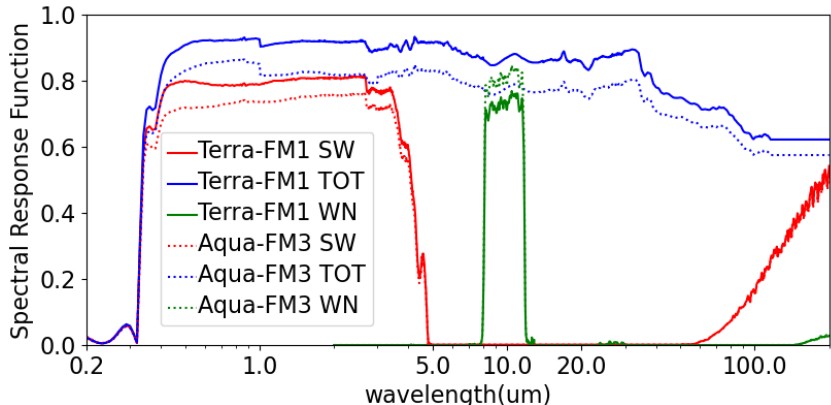


**Figure 1. Spectral response functions for CERES Terra FM1 and Aqua FM3 instruments.**


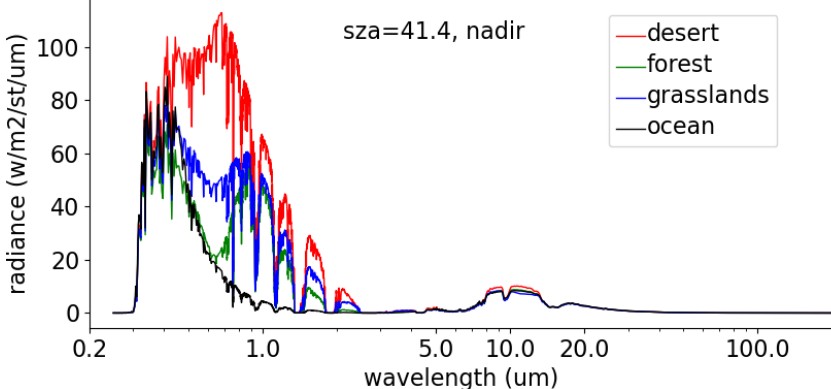

**Figure 2. MODTRAN 5.4 simulated clear-sky radiances at the top-of-atmosphere at SZA of 41.4° over desert, forest, grasslands and ocean surfaces with surface temperatures of 310, 300, 295, and 290 K, respectively.**


 



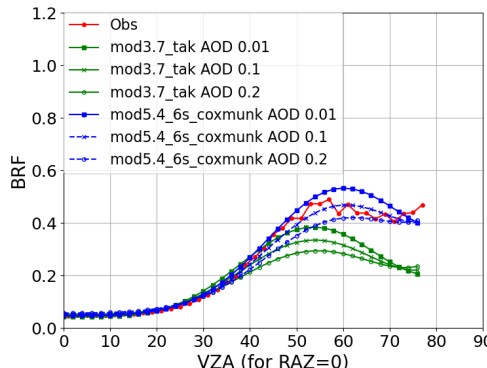

**Figure 3. Comparison of CERES Terra FM1 observed unfiltered BRFs in the solar principal plane at SZA of 49° to radiative transfer simulations with Cox-Munk ocean surface model (one uses implementation of Takashima (1985) with MODTRAN 3.7 and one uses implementation of the Satellite Signal in the Solar Spectrum (6S) radiative transfer code [Vermote et. al., 1997] with MODTRAN 5.4) with a wind speed of 5 m/s and a tropical profile. Three different AODs are shown for each model version.**


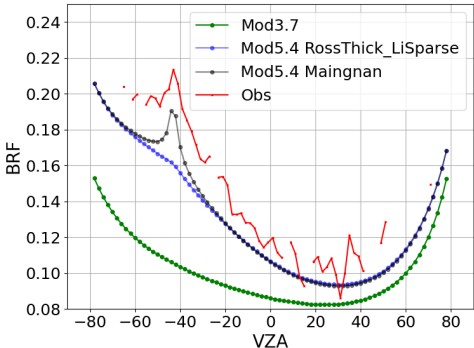

**Figure 4. Comparison of MODTRAN simulations to the CERES observed SW BRFs in the solar principal plane. The CERES observations are over the evergreen needleleaf forest in summer from 2000 to 2020 with SZA in the range of 40° and 42°. MODTRAN 3.7 simulations use Lambertian surface for coniferous forest described in Kriebel (1978). MODTRAN 5.4 simulations use the RossThick-LiSparse model and Maignan-modified RossThick-LiSparse model for the evergreen needleleaf forest with SZA of 41°.**







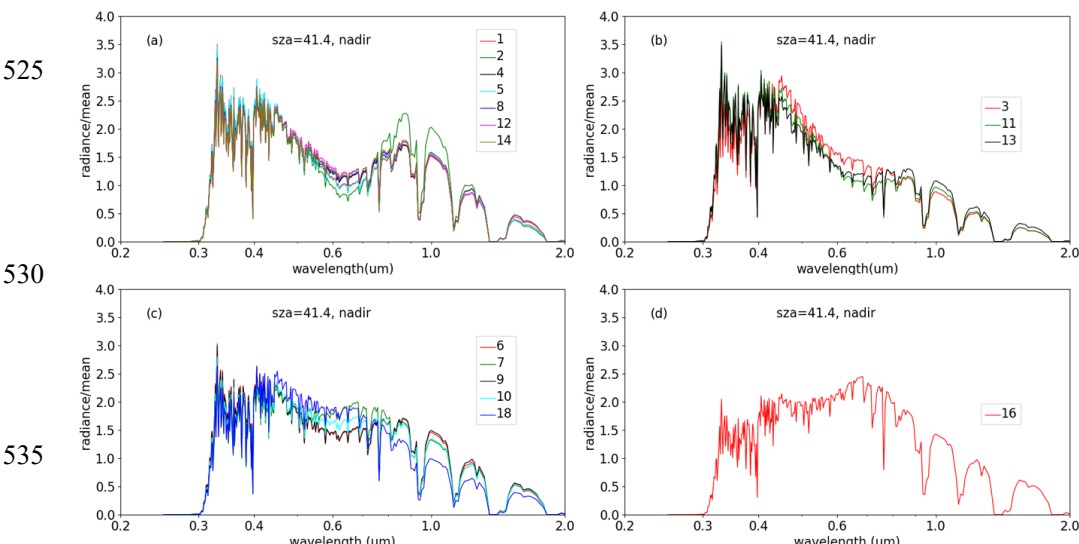

**Figure 5. MODTRAN 5.4 simulated clear-sky reflected radiances (normalized by the mean radiance across the spectrum) at the top-of-atmosphere at SZA of 41.4º for 16 land surface types defined by IGBP in January.**


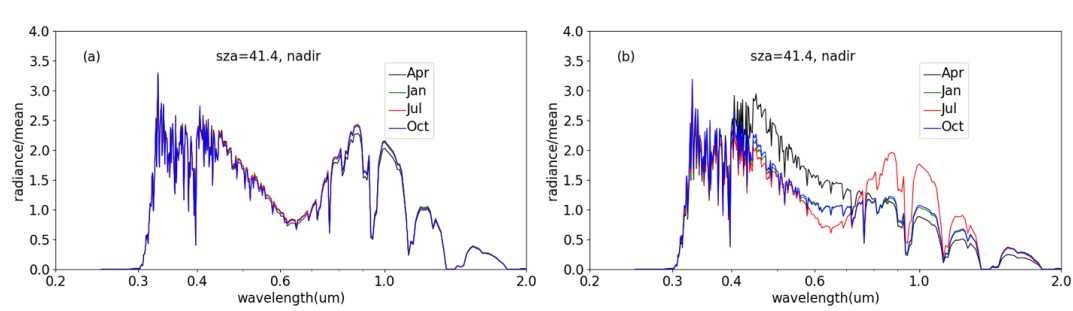

**Figure 6. MODTRAN 5.4 simulated clear-sky reflected radiances (normalized by the mean radiance across the spectrum) at the top-of-atmosphere at SZA of 41.4º for (a) evergreen broadleaf forest (IGBP=2) and (b) deciduous needleleaf forest (IGBP=3) in April, January, July, and October.**






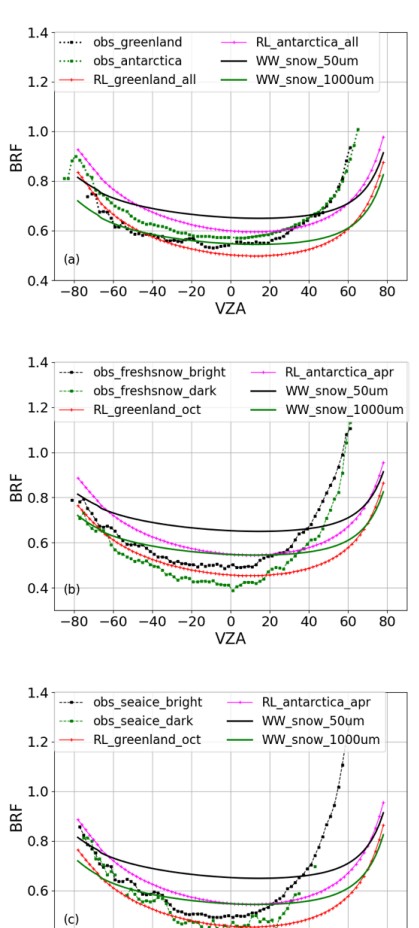

**Figure 7. (a) Comparison of MODTRAN 5.4 simulations with surface characterized by Warren-Wiscombe snow model**
**(snow grain size radius of 50 $\mu$m and 1000 $\mu$m) and RossThick-LiSparse kernel model to CERES observed BRFs for permanent snow in the solar principal plane at SZA of 75º. RL_greenland_all (RL_antarctica_all) stands for simulations using the averages of 10-year MODIS retrieved kernel parameters in all months over Greenland (Antarctica). (b) same as (a) but for fresh snow; RL_greenland_oct (RL_antarctica_apr) stands for simulations using the averages of 10-year MODIS retrieved kernel parameters in October over Greenland (Antarctica). Observed fresh snow BRFs are separated**
**into two categories (bright or dark) based on BRF magnitude values in nadir view. (c) same as (a) but for sea ice.**





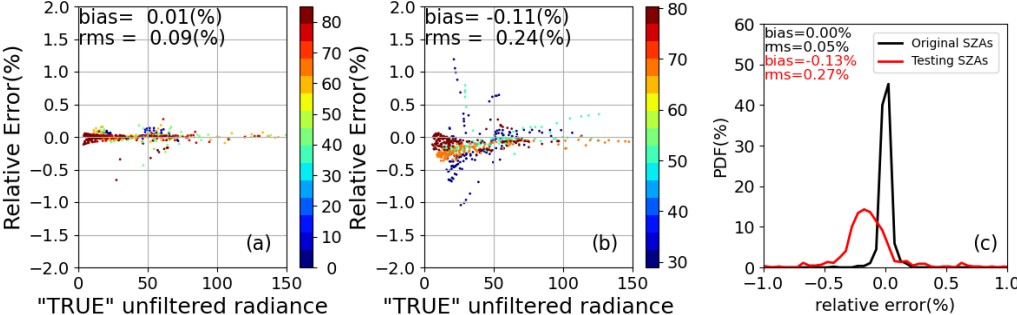


**Figure 8. SW unfiltered radiance errors by using unfiltering regression coefficients evaluated at 5 SZAs for clear-sky scenes over ocean. (a) Regression errors for SZAs at 0°, 41.4°, 60.0°, 75.5°, and 85.0°, at which the regression coefficients are evaluated in the Edition 4 CERES unfiltering process. (b) Errors estimated by alternative SZAs at 29.0°, 51.3°, 68.0°, and 80.3°. (c) Comparisons of the PDFs with data in the Left and Middle panels.**







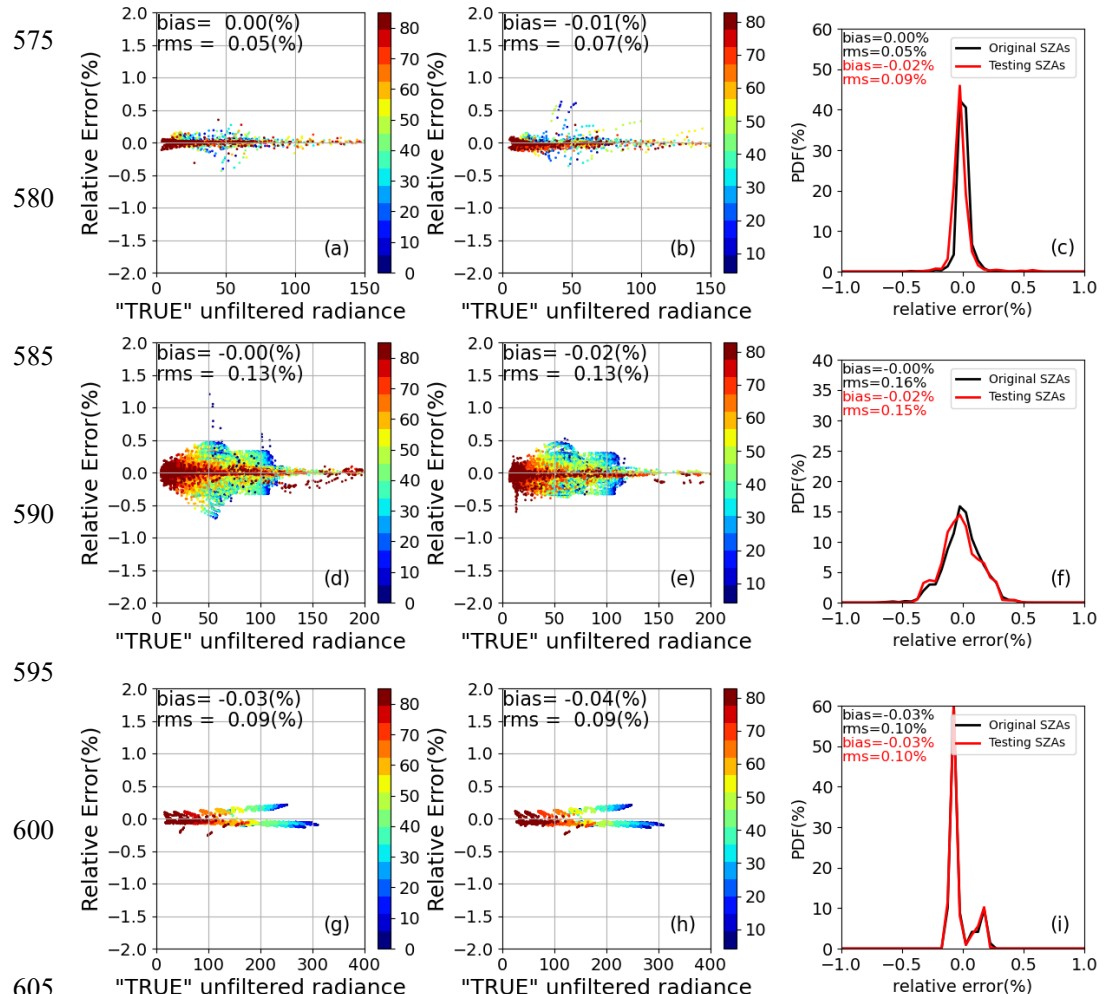

**Figure 9. Same as Figure 8, but by using unfiltering regression coefficients evaluated at 13 SZAs instead of 5 for scenes over ocean (a-c), land (d-f), and snow (g-i). The SZAs used to constructing regression coefficients are 0°, 8.3°, 16.6°, 23.6°, 29°, 35.7°, 41.4°, 51.3°, 60°, 68°, 75.5°, 80.3°, and 85° and the testing SZAs are 4.2°, 11.7°, 20.4°, 26.4°, 32.5°, 38.6°, 47°, 55.8°, 64.1°, 71.8°, 77.9°, and 82.7°.**






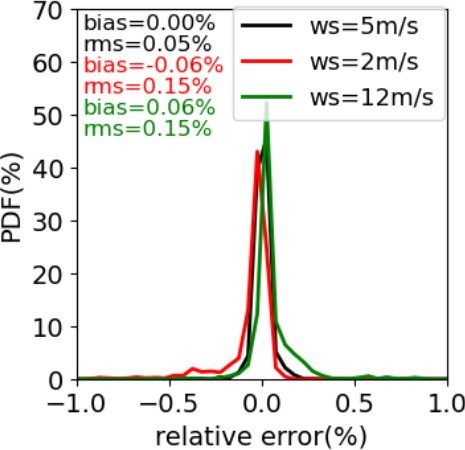


**Figure 10. SW unfiltered radiance errors for CERES Terra FM1 in clear-sky scenes over ocean with wind speeds of 2 m/s and 12 m/s as compared to that for simulations with a wind speed of 5 m/s, which is used to construct the regression coefficient for clear-sky scenes over ocean.**

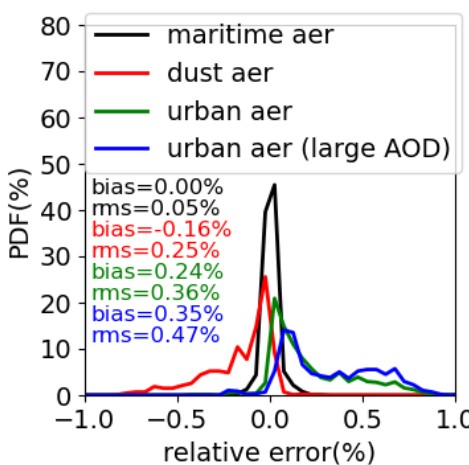

**Figure 11. SW unfiltered radiance errors for CERES Terra FM1 in clear-sky ocean scenes with dust, urban, and urban with relatively larger AOD aerosols as compared to that for simulations with maritime aerosols, which are used to construct regression coefficients for clear-sky scenes over ocean.**





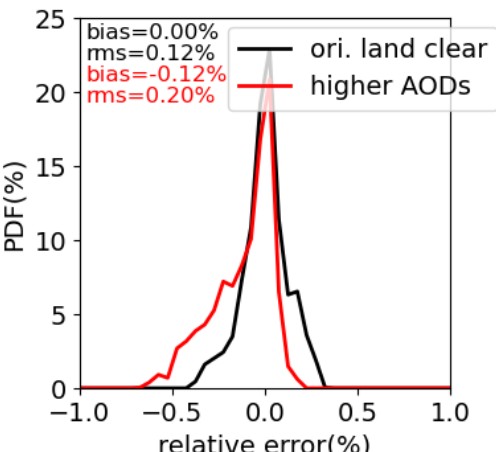

**Figure 12. SW unfiltered radiance errors for CERES Terra FM1 in clear-sky land scenes in July with large AODs (varying 630 from 0.5 to 2.0 depending on surface types) as compared to errors in regression coefficients derived from clear-sky land scenes, where AODs used in simulations vary from 0.05 to 0.83 depending on surface type.**

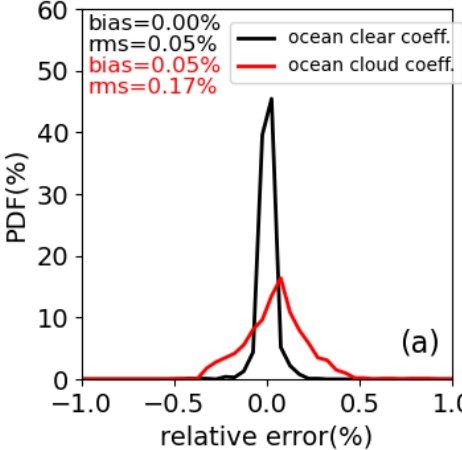
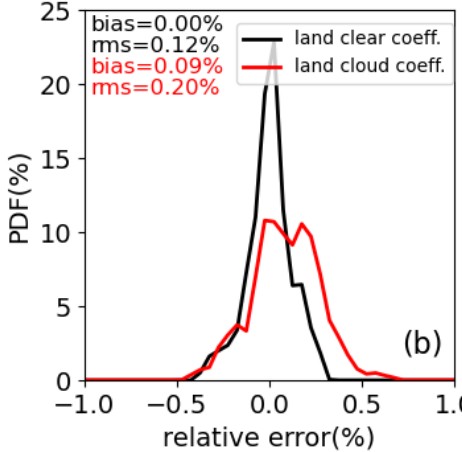

**Figure 13. Comparison of SW unfiltered radiance errors for clear-sky scenes by using the regression coefficients derived from cloudy sky and clear-sky scenes over ocean (a) and land in July (b) for CERES Terra FM1.**






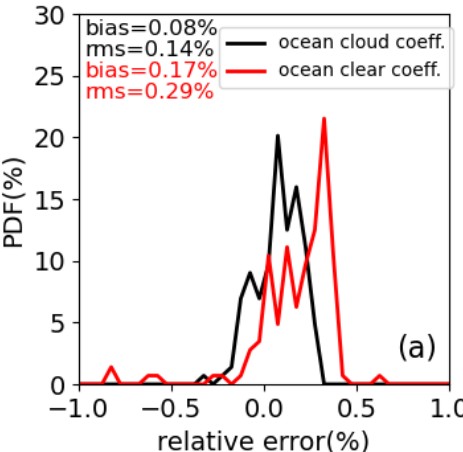
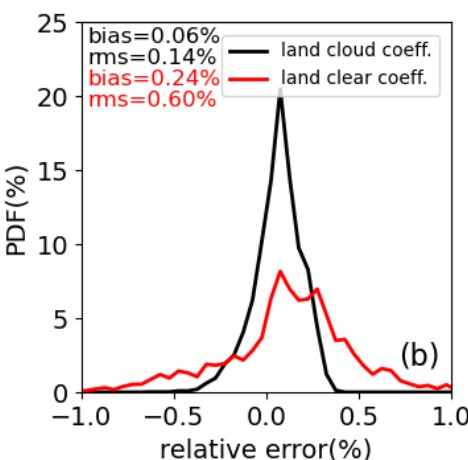

**Figure 14. Comparison of SW unfiltered radiance errors for overcast scenes covered by cirrus (COD=2) by using the regression coefficients derived from clear-sky scenes and cloudy sky scenes over ocean (a) and land in July (b) for CERES Terra FM1.**

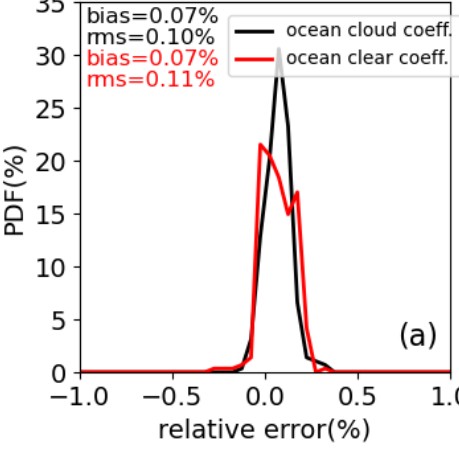
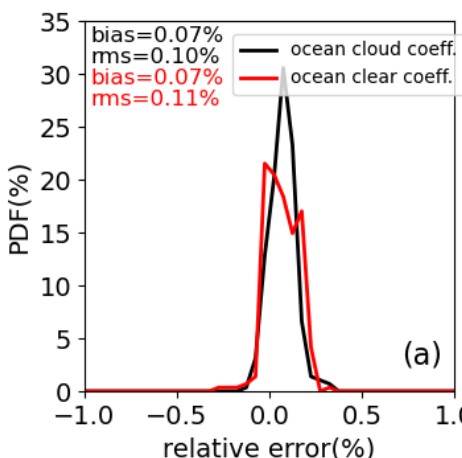

**Figure 15. Comparison SW unfiltered radiance errors for CERES Terra FM1 in broken cloudy-sky scenes (cirrus with COD=4 and stratus with COD=5.6 with a cloud fraction of 10%) by using the regression coefficients derived from cloudy sky scenes and clear-sky scenes over ocean (a) and land in July (b).**








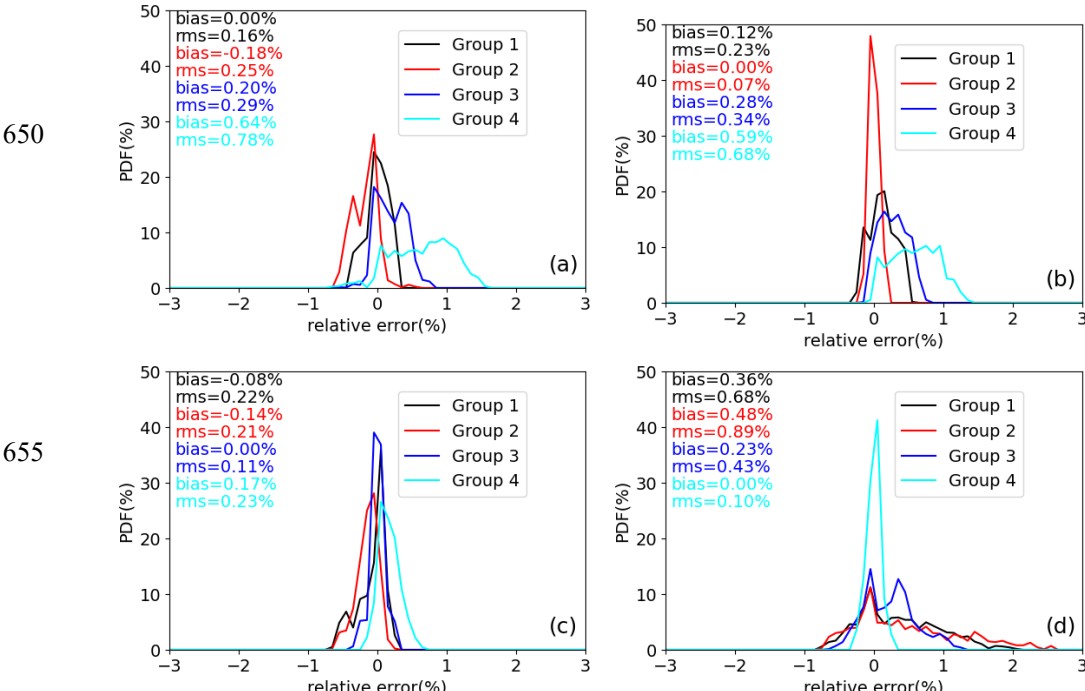

**Figure 16. SW unfiltered radiance errors for land surface (a) Group 1, (b) Group 2, (c) Group 3, and (d) Group 4 by using regression coefficients derived from all 4 land surface types, respectively, for CERES Terra FM1 in January. In January, Group 1 contains land surface IBGP types 01, 02, 04, 05, 08, 12 , and 14, Group 2 contains land IGBP surface types 03, 11, and 13, Group 3 contains land IGBP surface types 06, 07, 09, 10, and 18, and Group 4 contains land IGBP surface type 16.**






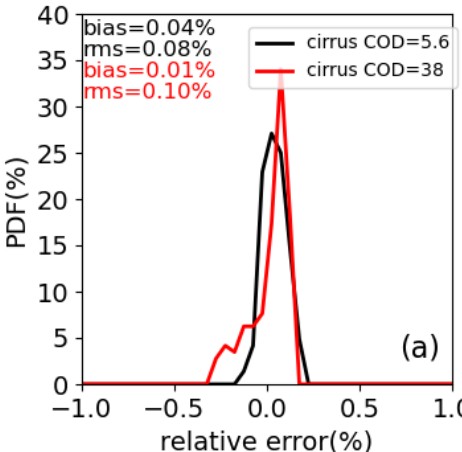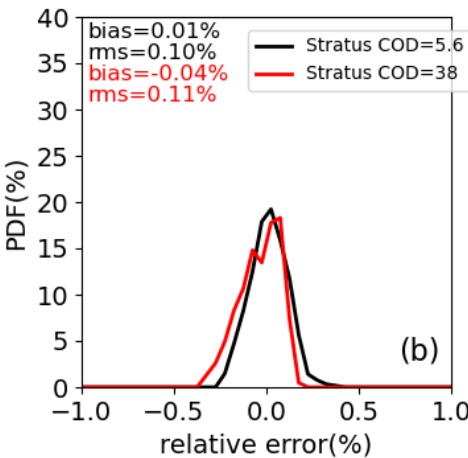

**Figure 17. SW unfiltered radiance errors for CERES Terra FM1 in cloudy sky scenes covered by stratus with COD of 38 as compared to that covered by stratus with COD of 5.6 by using regression coefficients derived from cloudy sky scenes over ocean (a) and land in July (b). Simulations with stratus of COD=5.6 are used to derive regression coefficients for cloudy sky scenes.**

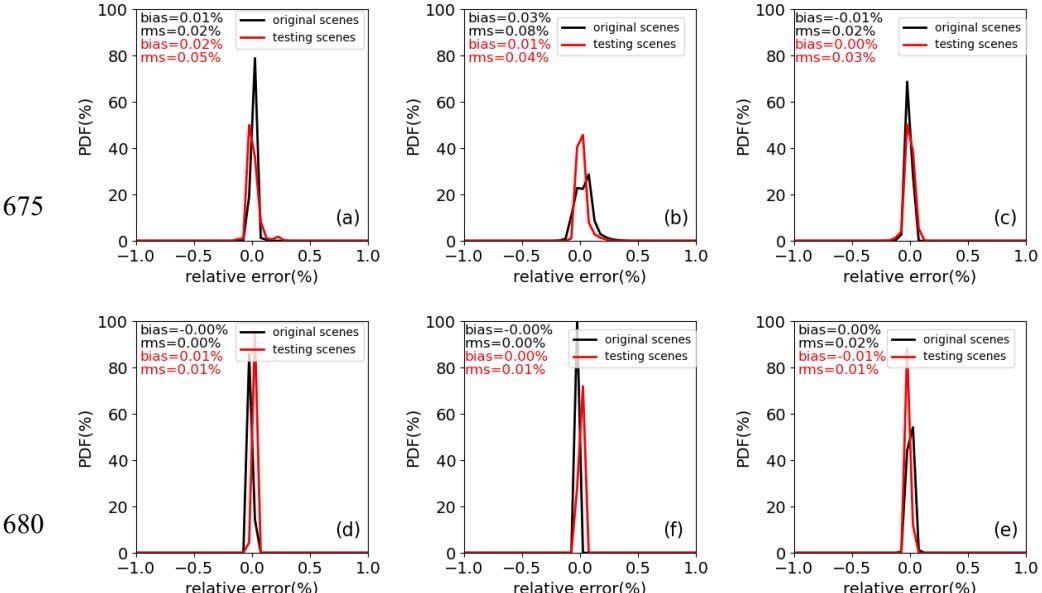

**Figure 18. Daytime (a-c) and nighttime (d-e) LW unfiltered radiance errors for CERES Terra FM1 in scenes with minimum and maximum climatological surface temperatures as compared to scenes with surface temperatures used in the simulations to derive the regression coefficients for ocean (a and d), land (d and f) and snow (c and e). For ocean, the minimum temperature is set to 260K; for snow, the maximum temperature is set to 273K.**

**Figure 19. Daytime (a-c) and nighttime (d-f) LW unfiltered radiance errors for CERES Terra FM1 in scenes with clouds placed at different altitudes as compared to overcast scenes used in simulations to derive the regression coefficients for cloudy sky over ocean (a and d), land (b and e) and snow (c and f).**

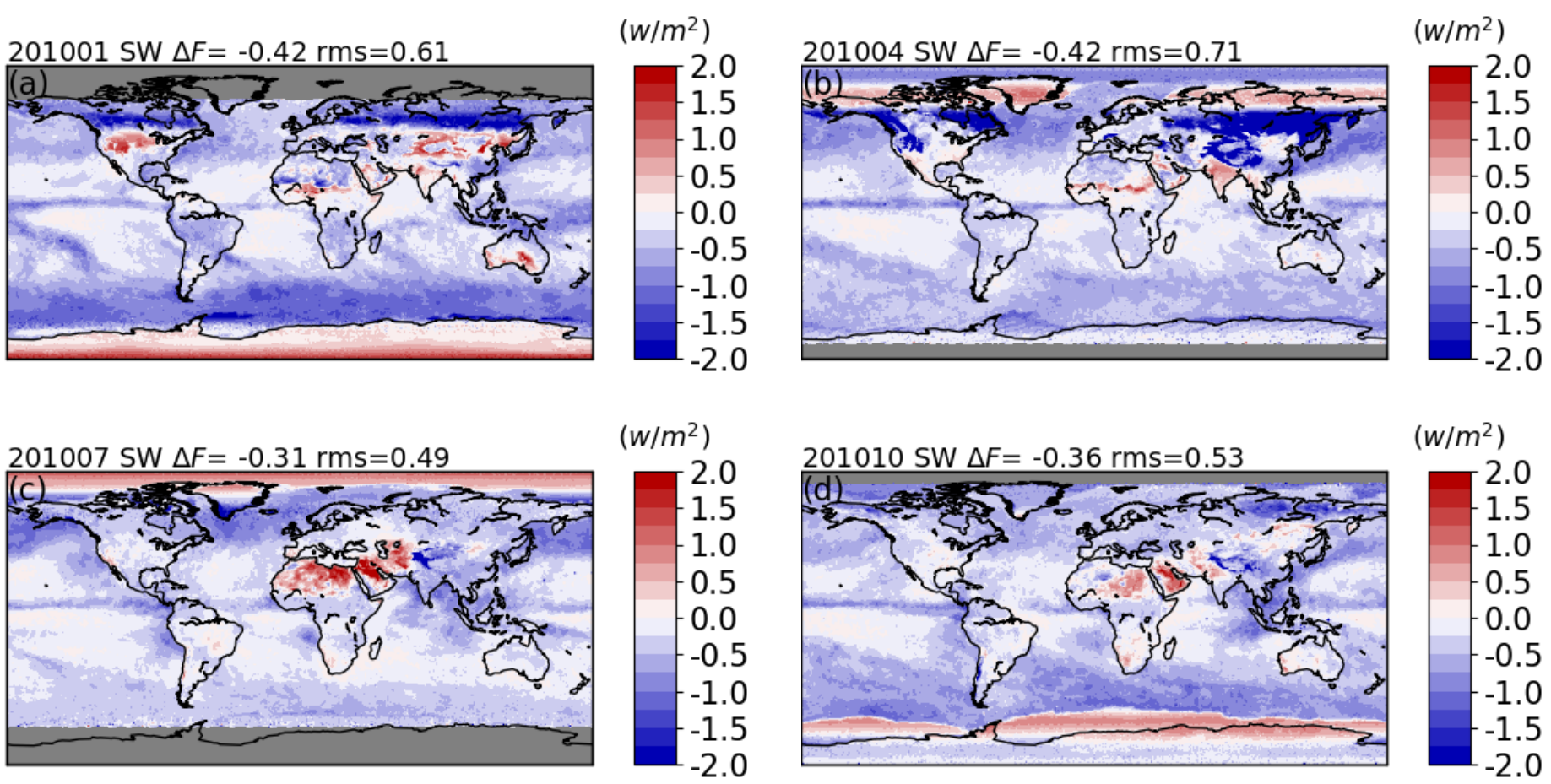

**Figure 20. The differences between the SW instantaneous fluxes retrieved with unfiltered radiances based on the updated CERES radiance unfiltering process and that based on the unfiltered radiances in the CERES Edition 4 product for the CERES Aqua FM3 instrument in April (a), July (b), October (c), and December (d).**






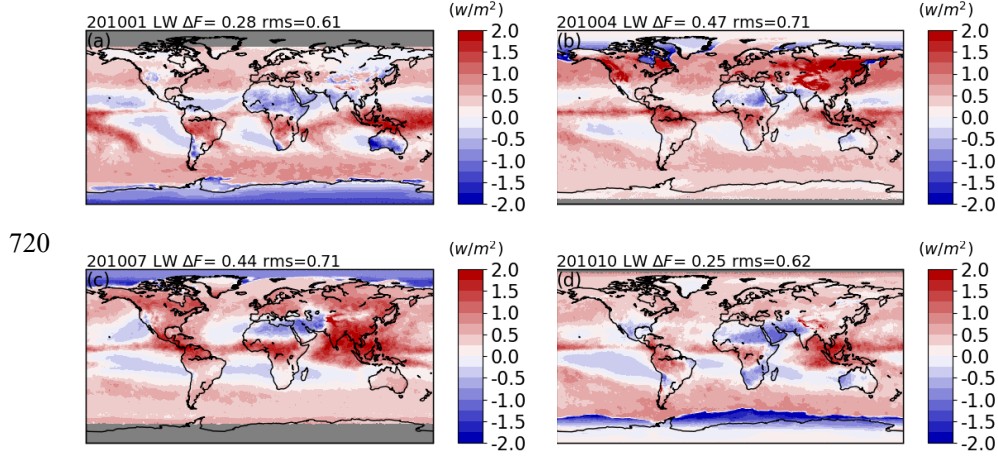

**Figure 21. Same as Figure 20, but for daytime LW.**


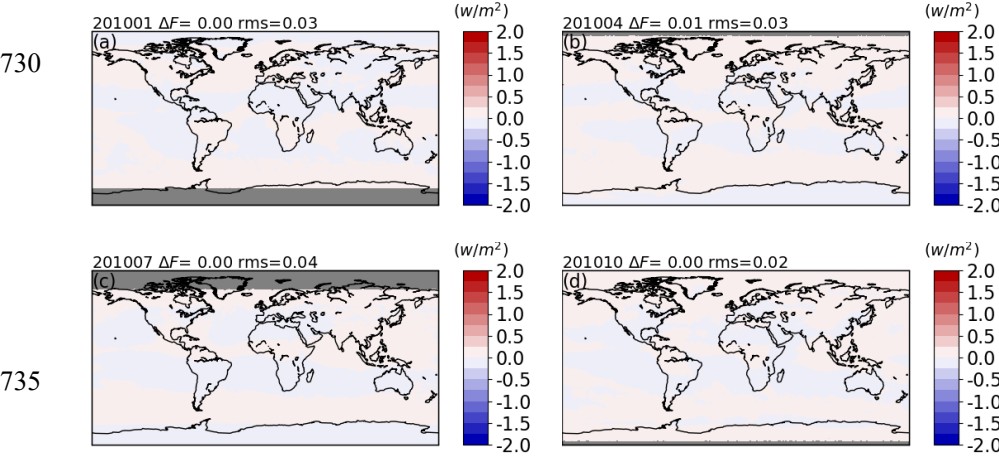


**Figure 22. Same as Figure 20, but for nighttime LW.**






**Table 1. Regression coefficients angular bin definitions**

| | |
|---|---|
| Solar zenith angle (°) | 0, 8.3, 16.6, 23.6, 29.0, 35.7, 41.4, 51.3, 60.0, 68.0, 75.5, 80.3, 85.0 |
| Viewing zenith angle(°) | 0, 30, 45, 60, 70, 90 |
| Relative azimuth angle(°) | 0, 7.5, 37.5, 90.0, 142.5, 172.5 |

**Table 2. Summary of cloud-free properties used in radiative transfer calculations for oceanic conditions with a wind speed of 5 m/s and a tropical atmospheric profile.**

| Aerosol type | Aerosol optical depth | Surface temperature (K) |
|---|---|---|
| --- | 0 | 320 |
| | 0.055 | 310 |
| | 0.090 | 300 |
| Maritime | 0.161 | 295 |
| | 0.301 | 290 |
| | 0.674 | 285 |
| | 1.171 | 280 |

**Table 3. Land surface type indices defined by International Geosphere-Biosphere Programme (IGBP) and the corresponding names.**

| | |
|---|---|
| 1 | Evergreen Needleleaf Forest |
| 2 | Evergreen Broadleaf Forest |
| 3 | Deciduous Needleleaf Forest |
| 4 | Deciduous Broadleaf Forest |
| 5 | Mixed Forest |
| 6 | Closed Shrublands |
| 7 | Open Shrublands |
| 8 | Woody Savannas |
| 9 | Savannas |
| 10 | Grasslands |
| 11 | Permanent Wetlands |
| 12 | Croplands |
| 13 | Urban and Built-up |
| 14 | Cropland Mosaics |
| 16 | Bare Soil and Rocks |
| 18 | Tundra |

**Table 4. Land surface type grouping in January, April, July, and October. The number(s) in a group is (are) IGBP surface type number(s).**

| month | Group 1 | Group 2 | Group 3 | Group 4 |
|---|---|---|---|---|
| January | 01, 02, 04, 05, 08, 12, 14 | 03, 11, 13 | 06, 07, 09, 10, 18 | 16 |
| April | 01, 02, 03, 04, 05, 08, 11, 12, 13, 14 | 06, 07, 09, 10 | 18 | 16 |
| July | 01, 02, 03, 04, 05, 06, 08, 11, 12, 14 | 07, 09, 10 | 13, 18 | 16 |
| October | 01, 03, 05, 11, 13, 18 | 02, 04, 06, 08, 09, 12, 14 | 07, 10 | 16 |



**Table 5. Summary of overcast properties used in radiative transfer calculations.**

| Surface | Clouds | Cloud optical depth (at 0.55 µm) | Cloud base height (km) | Cloud top height (km) |
|---|---|---|---|---|
| ocean and land | Ice | 4 | 9 | 10 |
| | | 12 | 9 | 11 |
| | Water | 5.6 | 0.5 | 0.7 |
| | | 217 | 0.66 | 2.9 |
| snow | Ice | 0.3 | 10 | 11 |
| | | 2 | 0.5 | 1.5 |
| | Water | 90 | 0.1 | 3.1 |
| | | 4 | 0.1 | 0.5 |

**Table 6. Summary of overcast deep convective cloud properties used in radiative transfer calculations.**

| Surface | Clouds | Cloud optical depth (at 0.55 um) | Cloud base height (km) | Cloud top height (km) |
|---|---|---|---|---|
| ocean | Ice | 210 | 5 | 12 |
| land | | 210 | 10 | 17 |

**Table 7. Angular bin definitions used to evaluate unfiltered radiance errors**

| | |
|---|---|
| Solar zenith angle (°) | 0, 29.0, 41.4, 51.3, 60.0, 68.0, 75.5, 80.3, 85.0 |
| Viewing zenith angle(°) | 0, 30, 45, 60, 70, 90 |
| Relative azimuth angle(°) | 0, 7.5, 37.5, 90.0, 142.5, 172.5 |

770

**Table 8. Summary of overcast properties used in radiative transfer calculations over ocean and land for the LW unfiltered radiance error analysis.**

| Surface | Clouds | Cloud optical depth (at 0.55 um) | Cloud base height | Cloud top height |
|---|---|---|---|---|
| ocean and land | ice | 4 | 7 | 8 |
| | | 4 | 10 | 11 |
| | | 12 | 7 | 9 |
| | | 12 | 10 | 12 |
| | water | 5.6 | 3.0 | 3.2 |
| | | 217 | 3 | 5.34 |
| snow | ice | 0.3 | 12 | 12.3 |
| | | 2 | 2.5 | 3.5 |
| | water | 90 | 0.5 | 3.5 |
| | | 4 | 0.5 | 0.9 |

775