# Peer review of "Next-generation radiance unfiltering process for the Clouds and Earth's Radiant Energy System instrument"

_EGUsphere, 2023_

## Author Comment (AC1)

This is overall an interesting paper that suggests many improvements to the NASA CERES unfiltering process. The error analysis is very relevant (section 3). Although this work could be a step toward implementation in future CERES releases (eg Ed5), it does not fully explain the differences wrt to Ed4 that are shown in section 4. Additional validation/verification/documentation would be welcome before implementation in the CERES processing system.

I am not a native speaker but get the feeling that the English for some sentences could be improved. A careful review, eg by the journal editor, is suggested. Similarly, the authors should check the units and symbols (eg micrometer is sometime written as µm, or um, or mm).

I suggest the following points for an improved manuscript:

- Don't use "less than" in the abstract and text body (eg line 23: "... are reduced by less than 0.31 W/m² ..." should be "... are reduced by 0.31 W/m² ...").

Changed as suggested.

- Typos line 16 ("process that used one set ..."), line 46 ("covering"), line 55 ("surface the surface"), line 91 ("relationships"), 132 ("µm").

Done.

- line 132, the wavelength range is given in µm and the spectral resolution in wavenumbers. It would be good to indicate how many wavelength steps have been used or to specify the wavelength increments, in µm, at lower end (0.25µm) and upper end (1000µm) of the wavelength range.

It is changed to the following:

Simulations are performed from 0 to 40000 wavenumbers per cm (0.25 µm to ~1000 µm) with a spectral resolution of 2 wavenumbers per cm.

- For the MODTRAN radiative transfer calculation in the LW part of the spectrum it would be interesting to specify how the surface emissivity has been considered (especially over desert surface).

In the Ed4 radiance unfiltering process, the land and snow/ice surfaces were characterized by Lambertian surfaces with prescribed spectral reflectances; in this version, the land and snow/ice surfaces are characterized by BRDF models and the emissivities are internally handled by MODTRAN through the surface BRDF reflectances.

- The handling of the far infra-red region should be discussed in more details. The spectral responses seem to be defined until 140µm (or 200 µm?), please confirm. What is the assumed sensitivity beyond this limit? zero? In this case, why are the MODTRAN simulations performed until 1000 µm.

The spectral response functions for SW and TOT channels are defined from 0.2 to 200 µm and they are 0 beyond these ranges. Although the simulations were prescribed up to 1000 µm, the simulated radiances beyond 200 µm are not used anyway.

- The FM1 and FM3 have marked difference in terms of spectral response (Figure 1). A brief discussion of the difference would be welcome. Also, the far-IR leakage of the SW filter seems to have an identical effect on the SW spectral response for FM1 and FM3. Please confirm as it seems strange to have difference in TW responses and not in SW in the far IR.

We like to discuss the spectral response function differences of FM1 and FM3, however and unfortunately, as how does the response functions changing as a function of spectrum impact the radiance unfiltering is not clear. Instead of speculations, we prefer to leave it as it is. It's true that SW channel has quite large response in the far-IR range, but the actual SW energy is quite small. We think that its impact to the radiance unfiltering is negligible.

- Figures 20-22 show 4 panels that are said to be for April (a), July (b), October (c) and December (d). This is visibly not the case (eg (a) should be Winter (December?)). Further, the text discusses the results for January (eg line 343). Please check and correct.

Corrected. They should be January (a), April (b), July (c), and October (d).

- The end-to-end sensitivity study of the unfiltering algorithm (section 4) is really interesting. Given the (significant) observed differences with the Ed4 fluxes, this sections would deserve a longer discussion as well on the methodology as on the interpretation of the observed differences.

We have added more discussion in this section and please see the text for the details.

- Comparing Figures 21 and 22, it seems that most of the daytime difference in LW flux is coming from the subtraction of the SW component in the TW channel. Please confirm this and consider performing additional studies to confirm this work is an improvement with respect to Ed4.

We agree that the regions with positive values in SW are more likely to have negative values in the daytime LW. This statement has been added in Section 4.

---

## Author Comment (AC2)

**Review of Next-generation radiance unfiltering process for the Clouds and Earth's Radiant Energy System instrument.**

The paper contains the details and error analysis of an updated unfiltering algorithm for the CERES SW and LW radiances. Uncertainties of the algorithm due to scene mis-identification, ocean wind speed differences and the number of angles used to determine the regressions are addressed by modelling. The effect of the update on the instantaneous fluxes compared to the existing algorithms is also demonstrated.

If these updates are expected to be incorporated into the next edition of the CERES data the information contained in the paper is of good significance, being of relevance to the community and within the scope of the AMT journal. The method makes use of a range of improvements over the existing algorithm and thus presents a clear advancement. The work is of good scientific quality and with some exceptions where more clarification is needed (see specific comments) the description of the method, error analysis and the conclusions reached are clear and valid.

The work is generally well presented and logically ordered, although needs to be edited for minor issues with English grammar throughout (some at noted in Editorial points at the end of this review but I this is not exhaustive).

**Specific comments by section:**

**Section 2: Methodology**:

The window channel of FM1 and FM3 and LW channel of FM6 is mentioned but unfiltering of these is not directly addressed. Are these calculations relevant to the window channel, if yes some mention of differences would be useful. If not, then maybe the window channel should not be equally emphasized in the methodology section, or it should be clearly stated that this update does not apply to this channel.

The unfiltered WN radiance errors are compared to Ed4 and they are also negligible. We have added this statement in the text.

The error analysis and flux differences for NPP FM5 and NOAA20 FM6 are briefly mentioned in the text and detailed results are available upon request.

It should also be clarified and made consistent in the remaining text that the unfiltering corrects for overlap between the emitted and reflected radiation streams. This is implied in the methodology but the other sections so of the paper speak specifically of SW and LW both filtered and unfiltered which were defined in terms of

the wavelength ranges of the channels in the introduction. A sentence at the end of the methodology to make clear than in the remainder of the paper when the SW and LW unfiltered are referred to they indicate the $m_u$ terms corresponding to the reflected and emitted radiation streams defined in this section should be sufficient.

A clarification is added in the end of methodology Section 2.1.

**Section 2.2: Spectral radiance simulations.**

The number of DISORT streams used for the calculation is not stated. Were tests/error analysis done to show this number was sufficient for all scenes at the angular resolution used particularly for highly anisotropic scenes such near glint angles and for larger particle ice cloud for example. This is a problem I have run into in similar simulations which can even with a relatively high number of streams can result in some unphysical results. It may be of interest to include a 'sufficient streams' section in the error analysis section.

In the MODTRAN simulations, the number of streams of DISORT is set to 8 for water clouds and 16 for ice clouds. This statement has been added in Section 2.2.6.

The following plots show the broadband radiances for water clouds with effective diameters of 20 μm and 40 μm, and for ice clouds with effective diameters of 46.34 μm and 140.15 μm at SZA=31 with COD=10. The forward scattering direction is to the left and the backward scattering direction is to the right. Based on the following examples, we believe that the number of streams is sufficient for our simulations.

[Figure]

Although comparisons with CERES observations are shown for snow, ocean and land surfaces to indicate the improvements of the new simulations, no example is shown for ice cloud. Rather in this case it is just stated (line 247) that 'As expected the new simulations are able to capture the cloud anistropic characteristics better', could the justification / evidence behind this statement be provided, is this form the references paper or other information?

For water clouds, it is obviously better to use cloud's microphysical properties calculated by the Mie theory, rather than the single scattering albedo and asymmetry factors in MODTRAN code. For ice clouds, same ice microphysical properties have been used throughout in the CERES products. A detailed comparison can be found in Loeb et al. (2018).

For deep convective cloud discussed in lines 250-251 is there some reference that can be given to support the values used as displayed in table 6?

The purpose of adding deep convective clouds is to envelop the radiance range for the unfiltering regressions. In the radiance unfiltering error analysis, we have tested various cloud top settings and found that the current values are optimal to yield the smallest unfiltering errors.

**Section 3: Error analysis**

The details and distributions of the test cases is a bit unclear in some cases. I think some more detail of this is needed to interpret the bias and rms of the error distribution associated with these cases. For example, for wind speed tests are angular resolution the same as the regression calculations, as wind speed effects are very angle specific is this sufficient to characterize the errors in use and does it include the specific combination of angles that are particularly sensitive to wind speed effects. Would errors be different if the angles were changed from those simulated along with the wind speed? Similar questions arise for the aerosol error analysis.

In Figure (a) below, we tested the unfiltered radiance errors in clear-sky scenes over ocean in the solar plane with wind speeds of 2, 5, and 12 m/s based on the unfiltering regression coefficients developed for 6 VZAs (0°, 30°, 45°, 60°, 70° and 90°). The largest errors are found around VZA of 10° in the backward direction. To mitigate the error, we decided to add another VZA at 15 to construct the unfiltering regression coefficients. Figure (b) shows that the errors are reduced, but the magnitude of errors can still be greater than 1.0%. This might be addressed in the future by adding more VZA bins for developing regression coefficients. This figure and discussion are added in the manuscript. With the modification, the general conclusions are not changed.

[Figure]

[Figure]

Figure. (a) SW unfiltered radiance errors for Terra FM1 in clear-sky scenes over ocean in the solar plane based on the radiance unfiltering coefficients developed for 6 VZA bins (VZA=0°, 30°, 45°, 60°, 70°, and 90°). The errors for scenes with wind speeds of 2 m/s and 12 m/s are compared to that for simulations with wind speed of 5 m/s, which is used to construct the regression coefficients for clear-sky scenes over ocean. (b) Same as (a) but for the errors based on the radiance unfiltering coefficients developed for 7 VZA bins (VZA=0°, 15°, 45°, 60°, 70°, and 90°).

For interpretation of the surface type grouping results it would be helpful to have some idea of the percentage of land surface in each group in each season.

Instead of adding statistics of the percentage of land surfaces in each group for a season, we will now use the following figure is to show how much the fluxes change if the unfiltering coefficients for a different land surface type are used. the figure and discussion are added in the manuscript.

[Figure]

Figure. Clear-sky land SW Flux differences between fluxes retrieved from the unfiltered radiances by using the unfiltering coefficients developed for (a) Group 1, (b) Group 2, (c) Group 3, and (d) Group 4, respectively, and that by using corresponding coefficients for each land surface type. Figures show the results for Aqua FM3 in January, 2010.

In any case although the analysis here concludes that the correct surface type should be used particularly in the case of bare soil and rocks, the effect of seasonality error

is not considered and the changing in grouping that occurs here which presumably is not as clear cut a transition as modelled.

The following figures show the flux differences in January if coefficients for July is used instead of using coefficients for January. We added this figure and discussion in Section 2.2.4.

[Figure]

Figure. (a) Land clear-sky SW Flux differences between fluxes retrieved from the unfiltered radiances based on the unfiltering coefficients for July and that for January. (b) same as (a) but for daytime LW. Figures show the results for Aqua FM3 in January, 2010.

Overall there seem to be a lot of plots associated with this section. A total 30 relative error distributions shown in Figures 8 to 19. Perhaps these results could be presented more succinctly and some of the individual plots moved to the supplementary material. For example in respect of figures 8 and 9, I think it is sufficient to state the results that when regressions are based on only 5 SZA unfiltering errors for intermediate SZA angles are seen to be significant larger than for the SZA on which the regression is based. This difference becomes insignificant when 13 SZAs are used for the regressions.

This suggestion was taken. Original Figure 9 has been removed.

If retained ,either in the main paper or supplementary material the relative error / true radiance scatterplots in figure 8 and 9 should specify what the colour scale is.

Color scale is for SZA. Fixed.

Also given the lack of structure and minimal variation seen in the 12 plots shown for the LW results (figure 18 and 19) it would again maybe be sufficient to state that differences between the original and test scenes were insignificant.

Original Figures 18 and 19 have been removed. Results are stated in the text.

Finally, it would be useful to put the errors in the context of the magnitude of the differences due to improvements and the stated errors of the existing CERES unfiltering both generally and for the cases known to be most problematic for the

existing algorithms (thin cirrus and cloud-free high concentration of submicron absorbing aerosol for the SW and very cold convective clouds for the LW). And summarize any remaining problematic scenes.

In this work, the errors are estimated with the simulations that are assumed to represent the truth. If there are simulations that can perfectly match the truth, we certainly would like to present how much of the errors will be reduced as compared to the errors presented in the Ed4 unfiltering process. Given that there are no such simulations available to draw a conclusion as to which version is better in terms of presented errors, the error estimates depend more on the extent to which they represent the truth of the chosen simulations. Given that the improvements of simulations in this version, we believe that the presented error estimates in this paper are more relevant.

**Section 4: Impact of the unfiltering on the instantaneous fluxes**

This section will be of particular interest to general users of the CERES products. I think that further expansion here would be welcome, putting the differences in the context of the existing uncertainties both from the unfiltering and from other terms. It would also be interesting to know if regional annual biases are likely reinforced or cancel in the annual mean and if there was significant interannual variability. The monthly variations would seem to indicate that this will vary with region/scene. Further discussion of the origin of the differences and seasonality would also be of interest, both in the context of the previously discussed errors in this unfiltering and in the context of the known limitations of the existing CERES unfiltering.

More discussion has been added in this section.

Line 344 states the differences can be greater than 2Wm-2 in magnitude, as the range of the plots only extends to 2Wm-2 it would be useful to know what this number is and where it occurred (I assume it is not significant enough to extend the range shown in the plots).

The flux difference PDFs show that there are 0.9%, 3.0%, 0.3%, 0.1% of 1°x1° grids in January, April, July, and October, respectively, that have the differences with a magnitude larger than 2 W/m$^2$; For LW, there are 0.1%, 1.7%, 0.4%, and 0.4% of 1°x1° grids that have the differences larger than 2 W/m$^2$. The text has been modified to reflect the above statement.

It is not essential, but it might be interesting to supplement the maps with a distribution plot in terms of percentage differences for more direct comparison to previous distributions.

Two figures for the flux relative differences are added.

**Section 5: Summary**

The paper has primarily discussed results relevant to the unfiltering of the LW and SW channels of FM1 and FM3. It would be helpful to include some discussion of the relevance of the results and error analysis to the newer FM5 and FM6.

Discussions are added.

Lines 376-380 as well as stating the 'mostly within' the specific cases where these norm errors are exceeded it elaborated on in terms of scenes and uncertainties as was done in Loeb 2001.

Compared to the uncertainties presented in Loeb et al. 2001, the uncertainties in this paper are analyzed with more cases and in greater depth. Because there are no simulations that perfectly match the true cases to be used to draw a conclusion as to which version is better in terms of presented uncertainties, the uncertainty estimates depend more on the extent to which they represent the truth of the chosen simulations. Given that the improvements of simulations in this version, we believe that the presented uncertainties in this paper are more relevant.

**Some Minor Editorial points**

Line 16 "It contrasts to the Edition 4 unfiltering processing that one set of regressions for land and snow, respectively" ->

"In contrast to the Edition 4 unfiltering process where one set of regions are used for land and snow respectively"

Fixed.

Line 24 "…increase to less than 0.47"  -> "increased to nearly 0.47"

Fixed.

Line 25 "though regions differences can be as large as 2.0" it is stated in the paper line 344 that they are more than 2.0 and in the summary line 385 'as large as 2.0" . Please decide if they are larger or as large and make this statement consistent thoughout.

More detailed descriptions are added in Section 4 as the percentage of 1°x1° grids where the magnitude of flux differences are greater than 2.0 W/m$^2$.

Line 43 "the reflected and emitted energy is spectrally different among different earth targets" à "the spectral distribution of reflected and emitted energy varies with earth target"

Fixed.

Line 170 "spectral radiation" -> "spectral radiation"

Fixied.

Line 193 "In this paper" -> specify if you mean in the update to be used in the new edition

Fixed.

Line 267 "..shows the error analysis based on .." -> "presents error analysis based on" ,

Fixed.

Line 271 "they are 0" -> "these are 0"

Fixed.

Line 271 " to evaluate if the 5 SZAs is sufficient, we used the simulations for clear sky with" -> "to evaluate if 5 SZAs are sufficient, we use clear sky simulation with"

Fixed.

Line 275 "For the number of VZA and RAZ, we verify that 6 VZA (increase one VZA at 70)..." -> "In respect of the number of the VZA and RAZ, it is verified that the increasing to 6 VZA by extending the VZA range to 70..."

Fixed.

Line 283 " It suggests that contructing .." -> "These results indicate ..."

Fixed.

Line 289 "...the PDFs of errors for other aerosols become broader, and the PDF modes for dust and urban aersols are shifted to opposite directions from one another..." -> "..the error PDFS for other aerosols are broader, and the mode of the PDF for dust and urban aerosols are shifted to negative and positive values respectively"

Fixed.

Line 291 ..”depending on the surface type..” -> add a clarification of this dependence.

Fixed.

Line 301 “…scenes by using the regressions..” -> “scenes unfiltered using the regionssions..”

Fixed.

Line 302 “The large PDF difference…” -> “The broader difference PDF..”

Fixed.

Line 317 “…”but larger otherwise.>” -à provide a more quantitative statement

Fixed.

Line 318 “..can easily be greter than 1%” -> quantify

Fixed.

Line 326 “Errors for LW” -> “Lw unfiltering errors” however as there hasn’t been a SW unfiltering section it would be consistent if all the previous sections didn’t pertin to the LW to make that clear earlier.

Changed the Section’s tile to “3.6 Errors due to surface and cloud top temperatures”

Line 348 “differences are positive…” -> include some quantification

Fixed.

Line 352 “nearly negligible” -> provide some maximum limit / quantification of negligible

Fixed.

Line 361 “with many updates compared to “ -> summarize updates or refer to section summarizing them

Fixed.

Line 364 "matches CERES observed SW radiances better' -> "matches the CERES observed angular variation of the SW radiances better"

Fixed.